# Deep Lagrangian Networks: Using Physics as Model Prior for Deep Learning

**Michael Lutter, Christian Ritter & Jan Peters** [*]
Department of Computer Science
Technische Universität Darmstadt
Hochschulstr. 10, 64289 Darmstadt, Germany
`{Lutter, Peters}@ias.tu-darmstadt.de`

## Abstract

Deep learning has achieved astonishing results on many tasks with large amounts of data and generalization within the proximity of training data. For many important real-world applications, these requirements are unfeasible and additional prior knowledge on the task domain is required to overcome the resulting problems. In particular, learning physics models for model-based control requires robust extrapolation from fewer samples – often collected online in real-time – and model errors may lead to drastic damages of the system.

Directly incorporating physical insight has enabled us to obtain a novel deep model learning approach that extrapolates well while requiring fewer samples. As a first example, we propose Deep Lagrangian Networks (DeLaN) as a deep network structure upon which Lagrangian Mechanics have been imposed. DeLaN can learn the equations of motion of a mechanical system (i.e., system dynamics) with a deep network efficiently while ensuring physical plausibility.

The resulting DeLaN network performs very well at robot tracking control. The proposed method did not only outperform previous model learning approaches at learning speed but exhibits substantially improved and more robust extrapolation to novel trajectories and learns online in real-time.

## 1 Introduction

In the last five years, deep learning has propelled most areas of learning forward at an impressive pace (Krizhevsky et al., 2012; Mnih et al., 2015; Silver et al., 2017) – with the exception of physically embodied systems. This lag in comparison to other application areas is somewhat surprising as learning physical models is critical for applications that control embodied systems, reason about prior actions or plan future actions (e.g., service robotics, industrial automation). Instead, most engineers prefer classical off-the-shelf modeling as it ensures physical plausibility – at a high cost of precise measurements[1] and engineering effort. These plausible representations are preferred as these models guarantee to extrapolate to new samples, while learned models only achieve good performance in the vicinity of the training data.

To learn a model that obtains physically plausible representations, we propose to use the insights from physics as a model prior for deep learning. In particular, the combination of deep learning and physics seems natural as the compositional structure of deep networks enables the efficient computation of the derivatives at machine precision (Raissi & Karniadakis, 2018) and, thus, can encode a differential equation describing physical processes. Therefore, we suggest to encode the physics prior in the form of a differential in the network topology. This adapted topology amplifies the information content of the training samples, regularizes the end-to-end training, and emphasizes robust models capable of extrapolating to new samples while simultaneously ensuring physical plausibility. Hereby, we concentrate on learning models of mechanical systems using the Euler-Lagrange-Equation, a second order ordinary differential equation (ODE) originating from Lagrangian Mechanics, as physics prior.

---

[*]Max Planck Institute for Intelligent Systems, Spemannstr. 41, 72076 Tübingen, Germany

[1]Highly precise models usually require taking the physical system apart and measuring the separated pieces (Albu-Schäffer, 2002).

We focus on learning models of mechanical systems as this problem is one of the fundamental challenges of robotics (de Wit et al., 2012; Schaal et al., 2002).

CONTRIBUTION

The contribution of this work is twofold. First, we derive a network topology called Deep Lagrangian Networks (DeLaN) encoding the Euler-Lagrange equation originating from Lagrangian Mechanics. This topology can be trained using standard end-to-end optimization techniques while maintaining physical plausibility. Therefore, the obtained model must comply with physics. Unlike previous approaches to learning physics (Atkeson et al., 1986; Ledezma & Haddadin, 2017), which engineered fixed features from physical assumptions requiring knowledge of the specific physical embodiment, we are 'only' enforcing physics upon a generic deep network. For DeLaN only the system state and the control signal are specific to the physical system but neither the proposed network structure nor the training procedure. Second, we extensively evaluate the proposed approach by using the model to control a simulated 2 degrees of freedom (dof) robot and the physical 7-dof robot Barrett WAM in real time. We demonstrate DeLaN's control performance where DeLaN learns the dynamics model online starting from random initialization. In comparison to analytic- and other learned models, DeLaN yields a better control performance while at the same time extrapolates to new desired trajectories.

In the following we provide an overview about related work (Section 2) and briefly summarize Lagrangian Mechanics (Section 3). Subsequently, we derive our proposed approach DeLaN and the necessary characteristics for end-to-end training are shown (Section 4). Finally, the experiments in Section 5 evaluate the model learning performance for both simulated and physical robots. Here, DeLaN outperforms existing approaches.

## 2   RELATED WORK

Models describing system dynamics, i.e. the coupling of control input $\boldsymbol{\tau}$ and system state $\mathbf{q}$, are essential for model-based control approaches (Ioannou & Sun, 1996). Depending on the control approach, the control law relies either on the forward model $f$, mapping from control input to the change of system state, or on the inverse model $f^{-1}$, mapping from system change to control input, i.e.,

$$f(\mathbf{q}, \dot{\mathbf{q}}, \boldsymbol{\tau}) = \ddot{\mathbf{q}}, \qquad f^{-1}(\mathbf{q}, \dot{\mathbf{q}}, \ddot{\mathbf{q}}) = \boldsymbol{\tau}. \qquad (1)$$

Examples for application of these models are inverse dynamics control (de Wit et al., 2012), which uses the inverse model to compensate system dynamics, while model-predictive control (Camacho & Alba, 2013) and optimal control (Zhou et al., 1996) use the forward model to plan the control input. These models can be either derived from physics or learned from data. The physics models must be derived for the individual system embodiment and requires precise knowledge of the physical properties (Albu-Schäffer, 2002). When learning the model[2], mostly standard machine learning techniques are applied to fit either the forward- or inverse-model to the training data. E.g., authors used Linear Regression (Schaal et al., 2002; Haruno et al., 2001), Gaussian Mixture Regression (Calinon et al., 2010; Khansari-Zadeh & Billard, 2011), Gaussian Process Regression (Kocijan et al., 2004; Nguyen-Tuong et al., 2009; Nguyen-Tuong & Peters, 2010), Support Vector Regression (Choi et al., 2007; Ferreira et al., 2007), feedforward- (Jansen, 1994; Lenz et al., 2015; Ledezma & Haddadin, 2017; Sanchez-Gonzalez et al., 2018) or recurrent neural networks (Rueckert et al., 2017) to fit the model to the observed measurements.

Only few approaches incorporate prior knowledge into the learning problem. Sanchez-Gonzalez et al. (2018) use the graph representation of the kinematic structure as input. While the work of Atkeson et al. (1986), commonly referenced as the standard system identification technique for robot manipulators (Siciliano & Khatib, 2016), uses the Newton-Euler formalism to derive physics features using the kinematic structure and the joint measurements such that the learning of the dynamics model simplifies to linear regression. Similarly, Ledezma & Haddadin (2017) hard-code these physics features within a neural network and learn the dynamics parameters using gradient descent rather than linear regression. Even though these physics features are derived from physics, the

---

[2]Further information can be found in the model learning survey by Nguyen-Tuong & Peters (2011).

learned parameters for mass, center of gravity and inertia must not necessarily comply with physics as the learned parameters may violate the positive definiteness of the inertia matrix or the parallel axis theorem (Ting et al., 2006). Furthermore, the linear regression is commonly underdetermined and only allows to infer linear combinations of the dynamics parameters and cannot be applied to close-loop kinematics (Siciliano & Khatib, 2016).

DeLaN follows the line of structured learning problems but in contrast to previous approaches guarantees physical plausibility and provides a more general formulation. This general formulation enables DeLaN to learn the dynamics for any kinematic structure, including kinematic trees and closed-loop kinematics, and in addition does not require any knowledge about the kinematic structure. Therefore, DeLaN is identical for all mechanical systems, which is in strong contrast to the Newton-Euler approaches, where the features are specific to the kinematic structure. Only the system state and input is specific to the system but neither the network topology nor the optimization procedure.

The combination of differential equations and Neural Networks has previously been investigated in literature. Early on Lagaris et al. (1998; 2000) proposed to learn the solution of partial differential equations (PDE) using neural networks and currently this topic is being rediscovered by Raissi & Karniadakis (2018); Sirignano & Spiliopoulos (2017); Long et al. (2017). Most research focuses on using machine learning to overcome the limitations of PDE solvers. E.g., Sirignano & Spiliopoulos (2017) proposed the Deep Galerkin method to solve a high-dimensional PDE from scattered data. Only the work of Raissi et al. (2017) took the opposite standpoint of using the knowledge of the specific differential equation to structure the learning problem and achieve lower sample complexity. In this paper, we follow the same motivation as Raissi et al. (2017) but take a different approach. Rather than explicitly solving the differential equation, DeLaN only uses the structure of the differential equation to guide the learning problem of inferring the equations of motion. Thereby the differential equation is only implicitly solved. In addition, the proposed approach uses different encoding of the partial derivatives, which achieves the efficient computation within a single feed-forward pass, enabling the application within control loops.

## 3    PRELIMINARIES: LAGRANGIAN MECHANICS

Describing the equations of motion for mechanical systems has been extensively studied and various formalisms to derive these equations exist. The most prominent are Newtonian-, Hamiltonian- and Lagrangian-Mechanics. Within this work Lagrangian Mechanics is used, more specifically the Euler-Lagrange formulation with non-conservative forces and generalized coordinates.[3] Generalized coordinates are coordinates that uniquely define the system configuration. This formalism defines the Lagrangian $L$ as a function of generalized coordinates $\mathbf{q}$ describing the complete dynamics of a given system. The Lagrangian is not unique and every $L$ which yields the correct equations of motion is valid. The Lagrangian is generally chosen to be

$$L = T - V \tag{2}$$

where $T$ is the kinetic energy and $V$ is the potential energy. The kinetic energy $T$ can be computed for all choices of generalized coordinates using $T = \frac{1}{2}\dot{\mathbf{q}}^T\mathbf{H}(\mathbf{q})\dot{\mathbf{q}}$, whereas $\mathbf{H}(\mathbf{q})$ is the symmetric and positive definite inertia matrix (de Wit et al., 2012). The positive definiteness ensures that all non-zero velocities lead to positive kinetic energy. Applying the calculus of variations yields the Euler-Lagrange equation with non-conservative forces described by

$$\frac{d}{dt}\frac{\partial L}{\partial \dot{\mathbf{q}}_i} - \frac{\partial L}{\partial \mathbf{q}_i} = \boldsymbol{\tau}_i \tag{3}$$

where $\boldsymbol{\tau}$ are generalized forces. Substituting $L$ and $dV/d\mathbf{q} = \mathbf{g}(\mathbf{q})$ into Equation 3 yields the second order ordinary differential equation (ODE) described by

$$\mathbf{H}(\mathbf{q})\ddot{\mathbf{q}} + \underbrace{\dot{\mathbf{H}}(\mathbf{q})\dot{\mathbf{q}} - \frac{1}{2}\left(\frac{\partial}{\partial \mathbf{q}}\left(\dot{\mathbf{q}}^T\mathbf{H}(\mathbf{q})\dot{\mathbf{q}}\right)\right)^T}_{:=\mathbf{c}(\mathbf{q},\dot{\mathbf{q}})} + \mathbf{g}(\mathbf{q}) = \boldsymbol{\tau} \tag{4}$$

---

[3]More information can be found in the textbooks (Greenwood, 2006; de Wit et al., 2012; Featherstone, 2007)

where $\mathbf{c}$ describes the forces generated by the Centripetal and Coriolis forces (Featherstone, 2007). Using this ODE any multi-particle mechanical system with holonomic constraints can be described. For example various authors used this ODE to manually derived the equations of motion for coupled pendulums (Greenwood, 2006), robotic manipulators with flexible joints (Book, 1984; Spong, 1987), parallel robots (Miller, 1992; Geng et al., 1992; Liu et al., 1993) or legged robots (Hemami & Wyman, 1979; Golliday & Hemami, 1977).

## 4 Incorporating Lagrangian Mechanics into Deep Learning

Starting from the Euler-Lagrange equation (Equation 4), traditional engineering approaches would estimate $\mathbf{H}(\mathbf{q})$ and $\mathbf{g}(\mathbf{q})$ from the approximated or measured masses, lengths and moments of inertia. On the contrary most traditional model learning approaches would ignore the structure and learn the inverse dynamics model directly from data. DeLaN bridges this gap by incorporating the structure introduced by the ODE into the learning problem and learns the parameters in an end-to-end fashion. More concretely, DeLaN approximates the inverse model by representing the unknown functions $\mathbf{g}(\mathbf{q})$ and $\mathbf{H}(\mathbf{q})$ as a feed-forward networks. Rather than representing $\mathbf{H}(\mathbf{q})$ directly, the lower-triangular matrix $\mathbf{L}(\mathbf{q})$ is represented as deep network. Therefore, $\mathbf{g}(\mathbf{q})$ and $\mathbf{H}(\mathbf{q})$ are described by

$$\hat{\mathbf{H}}(\mathbf{q}) = \hat{\mathbf{L}}(\mathbf{q}\,;\theta)\hat{\mathbf{L}}(\mathbf{q}\,;\theta)^T \qquad \hat{\mathbf{g}}(\mathbf{q}) = \hat{\mathbf{g}}(\mathbf{q}\,;\psi)$$

where $\hat{}$ refers to an approximation and $\theta$ and $\psi$ are the respective network parameters. The parameters $\theta$ and $\psi$ can be obtained by minimizing the violation of the physical law described by Lagrangian Mechanics. Therefore, the optimization problem is described by

$$(\theta^*, \psi^*) = \arg\min_{\theta,\psi} \; \ell\left(\hat{f}^{-1}(\mathbf{q}, \dot{\mathbf{q}}, \ddot{\mathbf{q}}\,;\theta,\psi),\; \boldsymbol{\tau}\right) \tag{5}$$

$$\text{with} \quad \hat{f}^{-1}(\mathbf{q}, \dot{\mathbf{q}}, \ddot{\mathbf{q}}\,;\theta, \psi) = \hat{\mathbf{L}}\hat{\mathbf{L}}^T \ddot{\mathbf{q}} + \frac{d}{dt}\left(\hat{\mathbf{L}}\hat{\mathbf{L}}^T\right)\dot{\mathbf{q}} - \frac{1}{2}\left(\frac{\partial}{\partial\mathbf{q}}\left(\dot{\mathbf{q}}^T\hat{\mathbf{L}}\hat{\mathbf{L}}^T\dot{\mathbf{q}}\right)\right)^T + \hat{\mathbf{g}} \tag{6}$$

$$\text{s.t.} \; 0 < \mathbf{x}^T\;\hat{\mathbf{L}}\hat{\mathbf{L}}^T\mathbf{x} \quad \forall\,\mathbf{x}\in\mathbb{R}_0^n \tag{7}$$

where $\hat{f}^{-1}$ is the inverse model and $\ell$ can be any differentiable loss function. The computational graph of $\hat{f}^{-1}$ is shown in Figure 1.

Using this formulation one can conclude further properties of the learned model. Neither $\hat{\mathbf{L}}$ nor $\hat{\mathbf{g}}$ are functions of $\dot{\mathbf{q}}$ or $\ddot{\mathbf{q}}$ and, hence, the obtained parameters should, within limits, generalize to arbitrary velocities and accelerations. In addition, the obtained model can be reformulated and used as a forward model. Solving Equation 6 for $\ddot{\mathbf{q}}$ yields the forward model described by

$$\hat{f}(\mathbf{q}, \dot{\mathbf{q}}, \boldsymbol{\tau}\,;\theta, \psi) = \left(\hat{\mathbf{L}}\hat{\mathbf{L}}^T\right)^{-1}\left(\boldsymbol{\tau} - \frac{d}{dt}\left(\hat{\mathbf{L}}\hat{\mathbf{L}}^T\right)\dot{\mathbf{q}} + \frac{1}{2}\left(\frac{\partial}{\partial\mathbf{q}}\left(\dot{\mathbf{q}}^T\hat{\mathbf{L}}\hat{\mathbf{L}}^T\dot{\mathbf{q}}\right)\right)^T - \hat{\mathbf{g}}\right) \tag{8}$$

where $\hat{\mathbf{L}}\hat{\mathbf{L}}^T$ is guaranteed to be invertible due to the positive definite constraint (Equation 7). However, solving the optimization problem of Equation 5 directly is not possible due to the ill-posedness of the Lagrangian $L$ not being unique. The Euler-Lagrange equation is invariant to linear transformation and, hence, the Lagrangian $L' = \alpha L + \beta$ solves the Euler-Lagrange equation if $\alpha$ is non-zero and $L$ is a valid Lagrangian. This problem can be mitigated by adding an additional penalty term to Equation 5 described by

$$(\theta^*, \psi^*) = \arg\min_{\theta,\psi} \; \ell\left(\hat{f}^{-1}(\mathbf{q}, \dot{\mathbf{q}}, \ddot{\mathbf{q}}\,;\theta,\psi),\; \boldsymbol{\tau}\right) + \lambda\,\Omega(\theta, \psi) \tag{9}$$

where $\Omega$ is the $L_2$-norm of the network weights.

Solving the optimization problem of Equation 9 with a gradient based end-to-end learning approach is non-trivial due to the positive definite constraint (Equation 7) and the derivatives contained in $\hat{f}^{-1}$. In particular, $d(\mathbf{L}\mathbf{L}^T)/dt$ and $\partial\left(\dot{\mathbf{q}}^T\mathbf{L}\mathbf{L}^T\dot{\mathbf{q}}\right)/\partial\mathbf{q}_i$ cannot be computed using automatic differentiation as $t$ is not an input of the network and most implementations of automatic differentiation do not allow the backpropagation of the gradient through the computed derivatives. Therefore, the derivatives contained in $\hat{f}^{-1}$ must be computed analytically to exploit the full gradient information for training

of the parameters. In the following we introduce a network structure that fulfills the positive-definite constraint for all parameters (Section 4.1), prove that the derivatives $d(\mathbf{L}\mathbf{L}^T)/dt$ and $\partial\left(\dot{\mathbf{q}}^T\mathbf{L}\mathbf{L}^T\dot{\mathbf{q}}\right)/\partial\mathbf{q}_i$ can be computed analytically (Section 4.2) and show an efficient implementation for computing the derivatives using a single feed-forward pass (Section 4.3). Using these three properties the resulting network architecture can be used within a real-time control loop and trained using standard end-to-end optimization techniques.

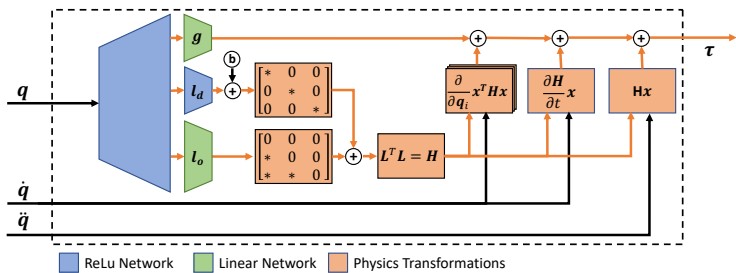

Figure 1: The computational graph of the Deep Lagrangian Network (DeLaN). Shown in blue and green is the neural network with the three separate heads computing $\mathbf{g}(\mathbf{q})$, $\mathbf{l}_d(\mathbf{q})$, $\mathbf{l}_o(\mathbf{q})$. The orange boxes correspond to the reshaping operations and the derivatives contained in the Euler-Lagrange equation. For training the gradients are backpropagated through all vertices highlighted in orange.

## 4.1 SYMMETRY AND POSITIVE DEFINITENESS OF $\mathbf{H}$

Ensuring the symmetry and positive definiteness of $\mathbf{H}$ is essential as this constraint enforces positive kinetic energy for all non-zero velocities. In addition, the positive definiteness ensures that $\mathbf{H}$ is invertible and the obtained model can be used as forward model. By representing the matrix $\mathbf{H}$ as the product of a lower-triangular matrix the symmetry and the positive semi-definiteness is ensured while simultaneously reducing the number of parameters. The positive definiteness is obtained if the diagonal of $\mathbf{L}$ is positive. This positive diagonal also guarantees that $\mathbf{L}$ is invertible. Using a deep network with different heads and altering the activation of the output layer one can obtain a positive diagonal. The off-diagonal elements $\mathbf{L}_o$ use a linear activation while the diagonal elements $\mathbf{L}_d$ use a non-negative activation, e.g., ReLu or Softplus. In addition, a positive scalar $b$ is added to diagonal elements. Thereby, ensuring a positive diagonal of $\mathbf{L}$ and the positive eigenvalues of $\mathbf{H}$. In addition, we chose to share parameters between $\mathbf{L}$ and $\mathbf{g}$ as both rely on the same physical embodiment. The network architecture, with three-heads representing the diagonal $\mathbf{l}_d$ and off-diagonal $\mathbf{l}_o$ entries of $\mathbf{L}$ and $\mathbf{g}$, is shown in Figure 1.

## 4.2 DERIVING THE DERIVATIVES

The derivatives $d\left(\mathbf{L}\mathbf{L}^T\right)/dt$ and $\partial\left(\dot{\mathbf{q}}^T\mathbf{L}\mathbf{L}^T\dot{\mathbf{q}}\right)/\partial\mathbf{q}_i$ are required for computing the control signal $\boldsymbol{\tau}$ using the inverse model and, hence, must be available within the forward pass. In addition, the second order derivatives, used within the backpropagation of the gradients, must exist to train the network using end-to-end training. To enable the computation of the second order derivatives using automatic differentiation the forward computation must be performed analytically. Both derivatives, $d\left(\mathbf{L}\mathbf{L}^T\right)/dt$ and $\partial\left(\dot{\mathbf{q}}^T\mathbf{L}\mathbf{L}^T\dot{\mathbf{q}}\right)/\partial\mathbf{q}_i$, have closed form solutions and can be derived by first computing the respective derivative of $\mathbf{L}$ and second substituting the reshaped derivative of the vectorized form $\mathbf{l}$. For the temporal derivative $d\left(\mathbf{L}\mathbf{L}^T\right)/dt$ this yields

$$\frac{d}{dt}\mathbf{H}(\mathbf{q}) = \frac{d}{dt}\left(\mathbf{L}\mathbf{L}^T\right) = \mathbf{L}\frac{d\mathbf{L}^T}{dt} + \frac{d\mathbf{L}}{dt}\mathbf{L}^T \tag{10}$$

whereas $d\mathbf{L}/dt$ can be substituted with the reshaped form of

$$\frac{d}{dt}\mathbf{l} = \frac{\partial\mathbf{l}}{\partial\mathbf{q}}\frac{\partial\mathbf{q}}{\partial t} + \sum_{i=1}^{N}\frac{\partial\mathbf{l}}{\partial\mathbf{W}_i}\frac{\partial\mathbf{W}_i}{\partial t} + \sum_{i=1}^{N}\frac{\partial\mathbf{l}}{\partial\mathbf{b}_i}\frac{\partial\mathbf{b}_i}{\partial t} \tag{11}$$

where $i$ refers to the $i$-th network layer consisting of an affine transformation and the non-linearity $g$, i.e., $\mathbf{h}_i = g_i\left(\mathbf{W}_i^T\mathbf{h}_{i-1} + \mathbf{b}_i\right)$. Equation 11 can be simplified as the network weights $\mathbf{W}_i$ and biases $\mathbf{b}_i$

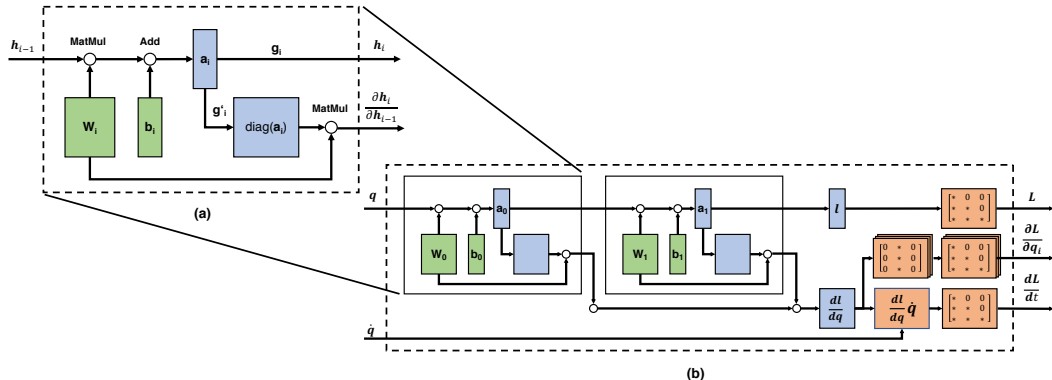

Figure 2: (a) Computational graph of the Lagrangian layer. The orange boxes highlight the learnable parameters. The upper computational sub-graph corresponds to the standard network layer while the lower sub-graph is the extension of the Lagrangian layer to simultaneously compute $\partial \mathbf{h}_i / \partial \mathbf{h}_{i-1}$. (b) Computational graph of the chained Lagrangian layer to compute $\mathbf{L}$, $d\mathbf{L}/dt$ and $\partial \mathbf{L}/\partial \mathbf{q}_i$ using a single feed-forward pass.

are time-invariant, i.e., $d\mathbf{W}_i/dt = 0$ and $d\mathbf{b}_i/dt = 0$. Therefore, $d\mathbf{l}/dt$ is described by

$$\frac{d}{dt}\mathbf{l} = \frac{\partial \mathbf{l}}{\partial \mathbf{q}}\dot{\mathbf{q}}. \tag{12}$$

Due to the compositional structure of the network and the differentiability of the non-linearity, the derivative with respect to the network input $d\mathbf{l}/d\mathbf{q}$ can be computed by recursively applying the chain rule, i.e.,

$$\frac{\partial \mathbf{l}}{\partial \mathbf{q}} = \frac{\partial \mathbf{l}}{\partial \mathbf{h}_{N-1}}\frac{\partial \mathbf{h}_{N-1}}{\partial \mathbf{h}_{N-2}} \cdots \frac{\partial \mathbf{h}_1}{\partial \mathbf{q}} \qquad \frac{\partial \mathbf{h}_i}{\partial \mathbf{h}_{i-1}} = \mathrm{diag}\left(g'(\mathbf{W}_i^T\mathbf{h}_{i-1} + \mathbf{b}_i)\right)\mathbf{W}_i \tag{13}$$

where $g'$ is the derivative of the non-linearity. Similarly to the previous derivation, the partial derivative of the quadratic term can be computed using the chain rule, which yields

$$\frac{\partial}{\partial \mathbf{q}_i}\left[\dot{\mathbf{q}}^T\mathbf{H}\dot{\mathbf{q}}\right] = \mathrm{tr}\left[\left(\dot{\mathbf{q}}\dot{\mathbf{q}}^T\right)^T\frac{\partial \mathbf{H}}{\partial \mathbf{q}_i}\right] = \dot{\mathbf{q}}^T\left(\frac{\partial \mathbf{L}}{\partial \mathbf{q}_i}\mathbf{L}^T + \mathbf{L}\frac{\partial \mathbf{L}}{\partial \mathbf{q}_i}^T\right)\dot{\mathbf{q}} \tag{14}$$

whereas $\partial \mathbf{L}/\partial \mathbf{q}_i$ can be constructed using the columns of previously derived $\partial \mathbf{l}/\partial \mathbf{q}$. Therefore, all derivatives included within $\hat{f}$ can be computed in closed form.

### 4.3 Computing the Derivatives

The derivatives of Section 4.2 must be computed within a real-time control loop and only add minimal computational complexity in order to not break the real-time constraint. $\mathbf{l}$ and $\partial \mathbf{l}/\partial \mathbf{q}$, required within Equation 10 and Equation 14, can be simultaneously computed using an extended standard layer. Extending the affine transformation and non-linearity of the standard layer with an additional sub-graph for computing $\partial \mathbf{h}_i / \partial \mathbf{h}_{i-1}$ yields the Lagrangian layer described by

$$\mathbf{a}_i = \mathbf{W}_i\mathbf{h}_{i-1} + \mathbf{b}_i \qquad \mathbf{h}_1 = g_i(\mathbf{a}_i) \qquad \frac{\partial \mathbf{h}_i}{\partial \mathbf{h}_{i-1}} = \mathrm{diag}\left(g_i'(\mathbf{a}_i)\right)\mathbf{W}_i.$$

The computational graph of the Lagrangian layer is shown in Figure 2a. Chaining the Lagrangian layer yields the compositional structure of $\partial \mathbf{l}/\partial \mathbf{q}$ (Equation 13) and enables the efficient computation of $\partial \mathbf{l}/\partial \mathbf{q}$. Additional reshaping operations compute $d\mathbf{L}/dt$ and $\partial \mathbf{L}/\partial \mathbf{q}_i$.

## 5 Experimental Evaluation: Learning an Inverse Dynamics Model for Robot Control

To demonstrate the applicability and extrapolation of DeLaN, the proposed network topology is applied to model-based control for a simulated 2-dof robot (Figure 3b) and the physical 7-dof robot

Figure 3: (a) Real-time control loop using a PD-Controller with a feed-forward torque $\tau_{FF}$, compensating the system dynamics, to control the joint torques $\tau$. The training process reads the joint states and applies torques to learn the system dynamics online. Once a new model becomes available the inverse model $\hat{f}^{-1}$ in the control loop is updated. (b) The simulated 2-dof robot drawing the cosine trajectories. (c) The simulated Barrett WAM drawing the 3d cosine 0 trajectory. (d) The physical Barrett WAM.

Barrett WAM (Figure 3d). The performance of DeLaN is evaluated using the tracking error on train and test trajectories and compared to a learned and analytic model. This evaluation scheme follows existing work (Nguyen-Tuong et al., 2009; Sanchez-Gonzalez et al., 2018) as the tracking error is the relevant performance indicator while the mean squared error (MSE)[4] obtained using sample based optimization exaggerates model performance (Hobbs & Hepenstal, 1989). In addition to most previous work, we strictly limit all model predictions to real-time and perform the learning online, i.e., the models are randomly initialized and must learn the model during the experiment.

EXPERIMENTAL SETUP

Within the experiment the robot executes multiple desired trajectories with specified joint positions, velocities and accelerations. The control signal, consisting of motor torques, is generated using a non-linear feedforward controller, i.e., a low gain PD-Controller augmented with a feed-forward torque $\tau_{ff}$ to compensate system dynamics. The control law is described by

$$\tau = \mathbf{K}_p(\mathbf{q}_d - \mathbf{q}) + \mathbf{K}_d(\dot{\mathbf{q}}_d - \dot{\mathbf{q}}) + \tau_{ff} \quad \text{with} \quad \tau_{ff} = \hat{f}^{-1}(\mathbf{q}_d, \dot{\mathbf{q}}_d, \ddot{\mathbf{q}}_d)$$

where $\mathbf{K}_p$, $\mathbf{K}_d$ are the controller gains and $\mathbf{q}_d$, $\dot{\mathbf{q}}_d$, $\ddot{\mathbf{q}}_d$ the desired joint positions, velocities and accelerations. The control-loop is shown in Figure 3a. For all experiments the control frequency is set to 500Hz while the desired joint state and respectively $\tau_{ff}$ is updated with a frequency of $f_d = 200$Hz. All feed-forward torques are computed online and, hence, the computation time is strictly limited to $T \leq 1/200$s. The tracking performance is defined as the sum of the MSE evaluated at the sampling points of the reference trajectory.

For the desired trajectories two different data sets are used. The first data set contains all single stroke characters[5] while the second data set uses cosine curves in joint space (Figure 3c). The 20 characters are spatially and temporally re-scaled to comply with the robot kinematics. The joint references are computed using the inverse kinematics. Due to the different characters, the desired trajectories contain smooth and sharp turns and cover a wide variety of different shapes but are limited to a small task space region. In contrast, the cosine trajectories are smooth but cover a large task space region.

BASELINES

The performance of DeLaN is compared to an analytic inverse dynamics model, a standard feedforward neural network (FF-NN) and a PD-Controller. For the analytic models the torque is computed using the Recursive Newton-Euler algorithm (RNE) (Luh et al., 1980), which computes the feedforward torque using estimated physical properties of the system, i.e. the link dimensions, masses and moments of inertia. For implementations the open-source library PyBullet (Coumans & Bai, 2016–2018) is used.

Both deep networks use the same dimensionality, ReLu nonlinearities and must learn the system dynamics online starting from random initialization. The training samples containing joint states and applied torques $(\mathbf{q}, \dot{\mathbf{q}}, \ddot{\mathbf{q}}, \tau)_{0...T}$ are directly read from the control loop as shown in Figure 3a.

[4]An offline comparisons evaluating the MSE on datasets can be found in the Appendix A.
[5]The data set was created by Williams et al. (2008) and is available at Dheeru & Karra Taniskidou (2017))

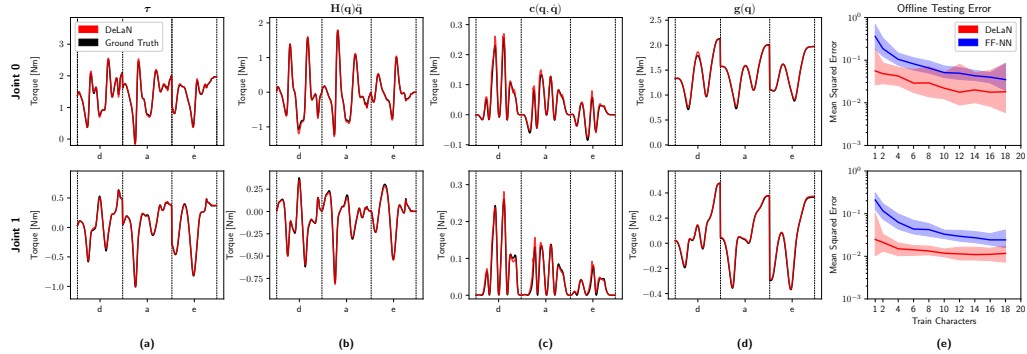

(a)      (b)      (c)      (d)      (e)

Figure 4: (a) The torque $\boldsymbol{\tau}$ required to generate the characters 'a', 'd' and 'e' in black. Using these samples DeLaN was trained offline and learns the red trajectory. DeLaN can not only learn the desired torques but also disambiguate the individual torque components even though DeLaN was trained on the super-imposed torques. Using Equation 6 DeLaN can represent the inertial force $\mathbf{H\ddot{q}}$ (b), the Coriolis and Centrifugal forces $\mathbf{c(q, \dot{q})}$ (c) and the gravitational force $\mathbf{g(q)}$ (d). All components match closely the ground truth data. (e) shows the offline MSE of the feed-forward neural network and DeLaN for each joint.

The training runs in a separate process on the same machine and solves the optimization problem online. Once the training process computed a new model, the inverse model $\hat{f}^{-1}$ of the control loop is updated.

## 5.1 SIMULATED ROBOT EXPERIMENTS

The 2-dof robot shown in Figure 3b is simulated using PyBullet and executes the character and cosine trajectories. Figure 4 shows the ground truth torques of the characters 'a', 'd', 'e', the torque ground truth components and the learned decomposition using DeLaN (Figure 4a-d). Even though DeLaN is trained on the super-imposed torques, DeLaN learns to disambiguate the inertial force $\mathbf{H\ddot{q}}$ , the Coriolis and Centrifugal force $\mathbf{c(q, \dot{q})}$ and the gravitational force $\mathbf{g(q)}$ as the respective curves overlap closely. Hence, DeLaN is capable of learning the underlying physical model using the proposed network topology trained with standard end-to-end optimization. Figure 4d shows the offline MSE on the test set averaged over multiple seeds for the FF-NN and DeLaN w.r.t. to different training set sizes. The different training set sizes correspond to the combination of *n* random characters, i.e., a training set size of 1 corresponds to training the model on a single character and evaluating the performance on the remaining 19 characters. DeLaN clearly obtains a lower test MSE compared to the FF-NN. Especially the difference in performance increases when the training set is reduced. This increasing difference on the test MSE highlights the reduced sample complexity and the good extrapolation to unseen samples. This difference in performance is amplified on the real-time control-task where the models are learned online starting from random initialization. Figure 5a and b shows the accumulated tracking error per testing character and the testing error averaged over all test characters while Figure 5c shows the qualitative comparison of the control performance[6]. It is important to point out that all shown results are averaged over multiple seeds and only incorporate characters not used for training and, hence, focus the evaluation on the extrapolation to new trajectories. The qualitative comparison shows that DeLaN is able to execute all 20 characters when trained on 8 random characters. The obtained tracking error is comparable to the analytic model, which in this case contains the simulation parameters and is optimal. In contrast, the FF-NN shows significant deviation from the desired trajectories when trained on 8 random characters. The quantitative comparison of the accumulated tracking error over seeds (Figure 5b) shows that DeLaN obtains lower tracking error on all training set sizes compared to the FF-NN. This good performance using only few training characters shows that DeLaN has a lower sample complexity and better extrapolation to unseen trajectories compared to the FF-NN.

Figure 6a and b show the performance on the cosine trajectories. For this experiment the models are only trained online on two trajectories with a velocity scale of 1x. To assess the extrapolation w.r.t. velocities and accelerations the learned models are tested on the same trajectories with scaled velocities (gray area of Figure 6). On the training trajectories DeLaN and the FF-NN perform

---

[6]The full results containing all characters are provided in the Appendix B.

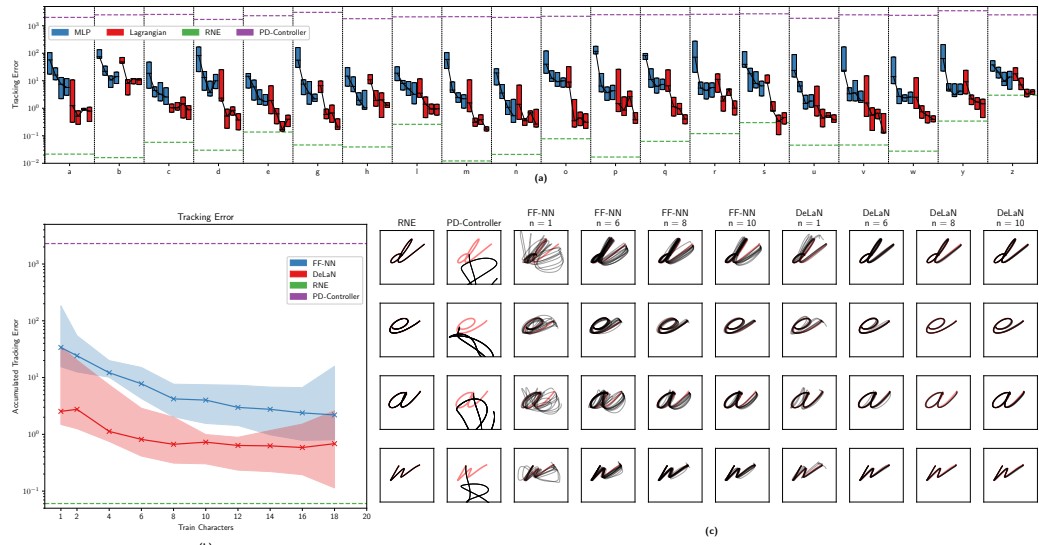

Figure 5: (a) The average performance of DeLaN and the feed forward neural network for each character. The 4 columns of the boxplots correspond to different numbers of training characters, i.e., $n = 1, 6, 8, 10$. (b) The median performance of DeLaN, the feed forward neural network and the analytic baselines averaged over multiple seeds. The shaded areas highlight the 5th and the 95th percentile. (c) The qualitative performance for the analytic baselines, the feed forward neural network and DeLaN. The desired trajectories are shown in red.

comparable. When the velocities are increased the performance of FF-NN deteriorates because the new trajectories do not lie within the vicinity of the training distribution as the domain of the FF-NN is defined as $(\mathbf{q}, \dot{\mathbf{q}}, \ddot{\mathbf{q}})$. Therefore, FF-NN cannot extrapolate to the testing data. In contrast, the domain of the networks $\hat{\mathbf{L}}$ and $\hat{\mathbf{g}}$ composing DeLaN only consist of $\mathbf{q}$, rather than $(\mathbf{q}, \dot{\mathbf{q}}, \ddot{\mathbf{q}})$. This reduced domain enables DeLaN, within limit, to extrapolate to the test trajectories. The increase in tracking error is caused by the structure of $\hat{f}^{-1}$, where model errors to scale quadratic with velocities. However, the obtained tracking error on the testing trajectories is significantly lower compared to FF-NN.

## 5.2 PHYSICAL ROBOT EXPERIMENTS

For physical experiments the desired trajectories are executed on the Barrett WAM, a robot with direct cable drives. The direct cable drives produce high torques generating fast and dexterous movements but yield complex dynamics, which cannot be modelled using rigid-body dynamics due to the variable stiffness and lengths of the cables[7]. Therefore, the Barrett WAM is ideal for testing the applicability of model learning and analytic models[8] on complex dynamics. For the physical experiments we focus on the cosine trajectories as these trajectories produce dynamic movements while character trajectories are mainly dominated by the gravitational forces. In addition, only the dynamics of the four lower joints are learned because these joints dominate the dynamics and the upper joints cannot be sufficiently excited to retrieve the dynamics parameters.

Figure 6c and d show the tracking error on the cosine trajectories using the the simulated Barrett WAM while Figure 6e and f show the tracking error of the physical Barrett WAM. It is important to note, that the simulation only simulates the rigid-body dynamics not including the direct cables drives and the simulation parameters are inconsistent with the parameters of the analytic model. Therefore, the analytic model is not optimal. On the training trajectories executed on the physical system the FF-NN performs better compared to DeLaN and the analytic model. DeLaN achieves slightly better tracking error than the analytic model, which uses the same rigid-body assumptions as DeLaN. That shows DeLaN can learn a dynamics model of the WAM but is limited by the model assumptions of Lagrangian Mechanics. These assumptions cannot represent the dynamics of the

---

[7]The cable drives and cables could be modelled simplistically using two joints connected by massless spring.
[8]The analytic model of the Barrett WAM is obtained using a publicly available URDF (JHU LCSR, 2018)

Figure 6: The tracking error of the cosine trajectories for the simulated 2-dof robot (a & b), the simulated (c & d) and the physical Barrett WAM (e & f). The feed-forward neural network and DeLaN are trained only on the trajectories at a velocity scale of 1×. Afterwards the models are tested on the same trajectories with increased velocities to evaluate the extrapolation to new velocities.

cable drives. When comparing to the simulated results, DeLaN and the FF-NN perform comparable but significantly better than the analytic model. These simulation results show that DeLaN can learn an accurate model of the WAM, when the underlying assumptions of the physics prior hold. The tracking performance on the physical system and the simulation indicate that DeLaN can learn a model within the model class of the physics prior but also inherits the limitations of the physics prior. For this specific experiment the FF-NN can locally learn correlations of the torques w.r.t. $\mathbf{q}$, $\dot{\mathbf{q}}$ and $\ddot{\mathbf{q}}$ while such correlation cannot be represented by the network topology of DeLaN because such correlation should, by definition of the physics prior, not exist.

When extrapolating to the identical trajectories with higher velocities (gray area of Figure 6) the tracking error of the FF-NN deteriorates much faster compared to DeLaN, because the FF-NN overfits to the training data. The tracking error of the analytic model remains constant and demonstrates the guaranteed extrapolation of the analytic models. When comparing the simulated results, the FF-NN cannot extrapolate to the new velocities and the tracking error deteriorates similarly to the performance on the physical robot. In contrast to the FF-NN, DeLaN can extrapolate to the higher velocities and maintains a good tracking error. Even further, DeLaN obtains a better tracking error compared the analytic model on all velocity scales. This low tracking error on all test trajectories highlights the improved extrapolation of DeLaN compared to other model learning approaches.

## 6 CONCLUSION

We introduced the concept of incorporating a physics prior within the deep learning framework to achieve lower sample complexity and better extrapolation. In particular, we proposed Deep Lagrangian Networks (DeLaN), a deep network on which Lagrangian Mechanics is imposed. This specific network topology enabled us to learn the system dynamics using end-to-end training while maintaining physical plausibility. We showed that DeLaN is able to learn the underlying physics from a super-imposed signal, as DeLaN can recover the contribution of the inertial-, gravitational and centripetal forces from sensor data. The quantitative evaluation within a real-time control loop assessing the tracking error showed that DeLaN can learn the system dynamics online, obtains lower sample complexity and better generalization compared to a feed-forward neural network. DeLaN can extrapolate to new trajectories as well as to increased velocities, where the performance of the feed-forward network deteriorates due to the overfitting to the training data. When applied to a physical systems with complex dynamics the bounded representational power of the physics prior can be limiting. However, this limited representational power enforces the physical plausibility and obtains the lower sample complexity and substantially better generalization. In future work the physics prior should be extended to represent a wider system class by introducing additional non-conservative forces within the Lagrangian.

ACKNOWLEDGMENTS

This project has received funding from the European Union's Horizon 2020 research and innovation program under grant agreement No #640554 (SKILLS4ROBOTS). Furthermore, this research was also supported by grants from ABB, NVIDIA and the NVIDIA DGX Station.

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

## Appendix A: Offline Benchmarks

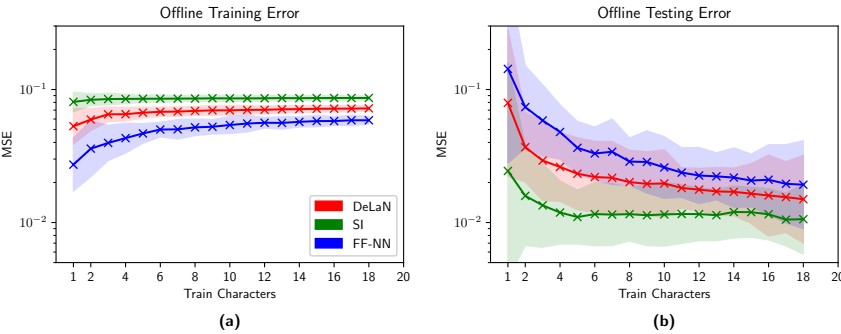

Figure 7: The mean squared error averaged of 20 seeds on the training- (a) and test-set (b) of the character trajectories for the two joint robot. The models are trained offline using $n$ characters and tested using the remaining $20 - n$ characters. The training samples are corrupted with white noise, while the performance is tested on noise-free trajectories.

To evaluate the performance of DeLaN without the control task, DeLaN was trained offline on previously collected data and evaluated using the mean squared error (MSE) on the test and training set. For comparison, DeLaN is compared to the system identification approach (SI) described by Atkeson et al. (1986), a feed-forward neural network (FF-NN) and the Recursive Newton Euler algorithm (RNE) using an analytic model. For this comparison, one must point out that the system identification approach relies on the availability of the kinematics, as the Jacobians and transformations w.r.t. to every link must be known to compute the necessary features. In contrast, neither DeLaN nor the FF-NN require this knowledge and must implicitly also learn the kinematics.

Figure 7 shows the MSE averaged over 20 seeds on the character data set executed on the two-joint robot. For this data set, the models are trained using noisy samples and evaluated on the noise-free and previously unseen characters. The FF-NN performs the best on the training set, but overfits to the training data. Therefore, the FF-NN does not generalize to unseen characters. In contrast, the SI approach does not overfit to the noise and extrapolates to previously unseen characters. In comparison, the structure of DeLaN regularizes the training and prevents the overfitting to the corrupted training data. Therefore, DeLaN extrapolates better than the FF-NN but not as good as the SI approach. Similar results can be observed on the cosine data set using the Barrett WAM simulated in SL (Figure 8 a, b). The FF-NN performs best on the training trajectory but the performance deteriorates when this network extrapolates to higher velocities. SI performs worse on the training trajectory but extrapolates to higher velocities. In comparison, DeLaN performs comparable to the SI approach on the training trajectory, extrapolates significantly better than the FF-NN but does not extrapolate as good as the SI approach. For the physical system (Figure 8 c, d), the results differ from the results in simulation. On the physical system the SI approach only achieves the same performance as RNE, which is significantly worse compared to the performance of DeLaN and the FF-NN. When evaluating the extrapolation to higher velocities, the analytic model and the SI approach extrapolate to higher velocities, while the MSE for the FF-NN significantly increases. In comparison, DeLaN extrapolates better compared to the FF-NN but not as good as the analytic model or the SI approach.

This performance difference between the simulation and physical system can be explained by the underlying model assumptions and the robustness to noise. While DeLaN only assumes rigid-body dynamics, the SI approach also assumes the exact knowledge of the kinematic structure. For simulation both assumptions are valid. However, for the physical system, the exact kinematics are unknown due to production imperfections and the direct cable drives applying torques to flexible joints violate the rigid-body assumption. Therefore, the SI approach performs significantly worse on the physical system. Furthermore, the noise robustness becomes more important for the physical system due to the inherent sensor noise. While the linear regression of the SI approach is easily corrupted by noise or outliers, the gradient based optimization of the networks is more robust to noise. This robustness can be observed in Figure 9, which shows the correlation between the variance of Gaussian noise corrupting the training data and the MSE of the simulated and noise-free cosine

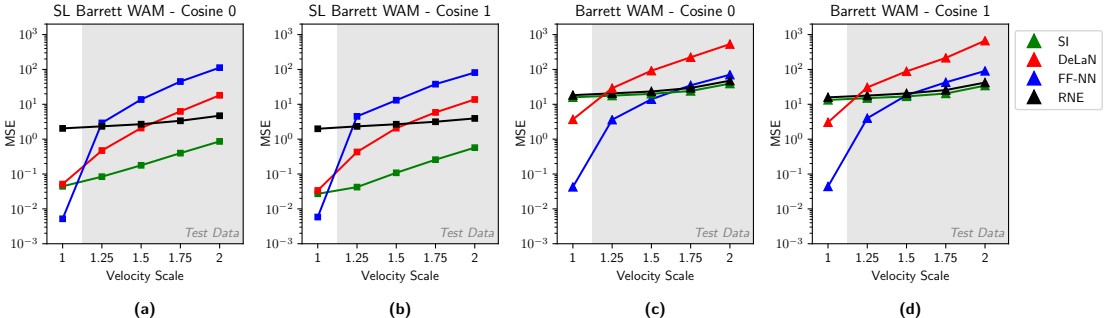

Figure 8: The mean squared error of the cosine trajectories for the simulated (a, b) and the physical Barrett WAM (c and d). The system identification approach, feed-forward neural network and DeLaN are trained offline using only the trajectories at a velocity scale of 1×. Afterwards the models are tested on the same trajectories with increased velocities to evaluate the extrapolation to new velocities.

trajectories. With increasing noise levels, the MSE of the SI approach increases significantly faster compared to the models learned using gradient descent.

Concluding, the extrapolation of DeLaN to unseen trajectories and higher velocities is not as good as the SI approach but significantly better than the generic FF-NN. This increased extrapolation compared to the generic network is achieved by the Lagrangian Mechanics prior of DeLaN. Even though this prior promotes extrapolation, the prior also hinders the performance on the physical robot, because the prior cannot represent the dynamics of the direct cable drives. Therefore, DeLaN performs worse than the FF-NN, which does not assume any model structure. However, DeLaN outperforms the SI approach on the physical system, which also assumes rigid-body dynamics and requires the exact knowledge of the kinematics.

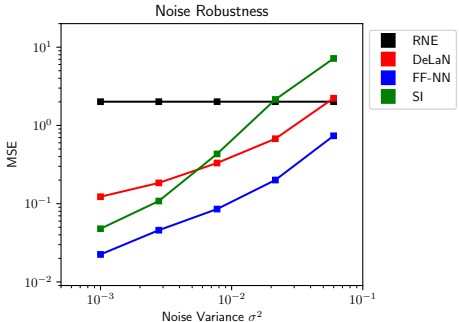

Figure 9: The mean squared error on the simulated and noise-free cosine trajectories with velocity scale of 1x. For offline training the samples are corrupted using i.i.d. noise sampled from a multivariate Normal distribution with the variance of $\sigma^2 \mathbf{I}$.

APPENDIX B: COMPLETE ONLINE RESULTS

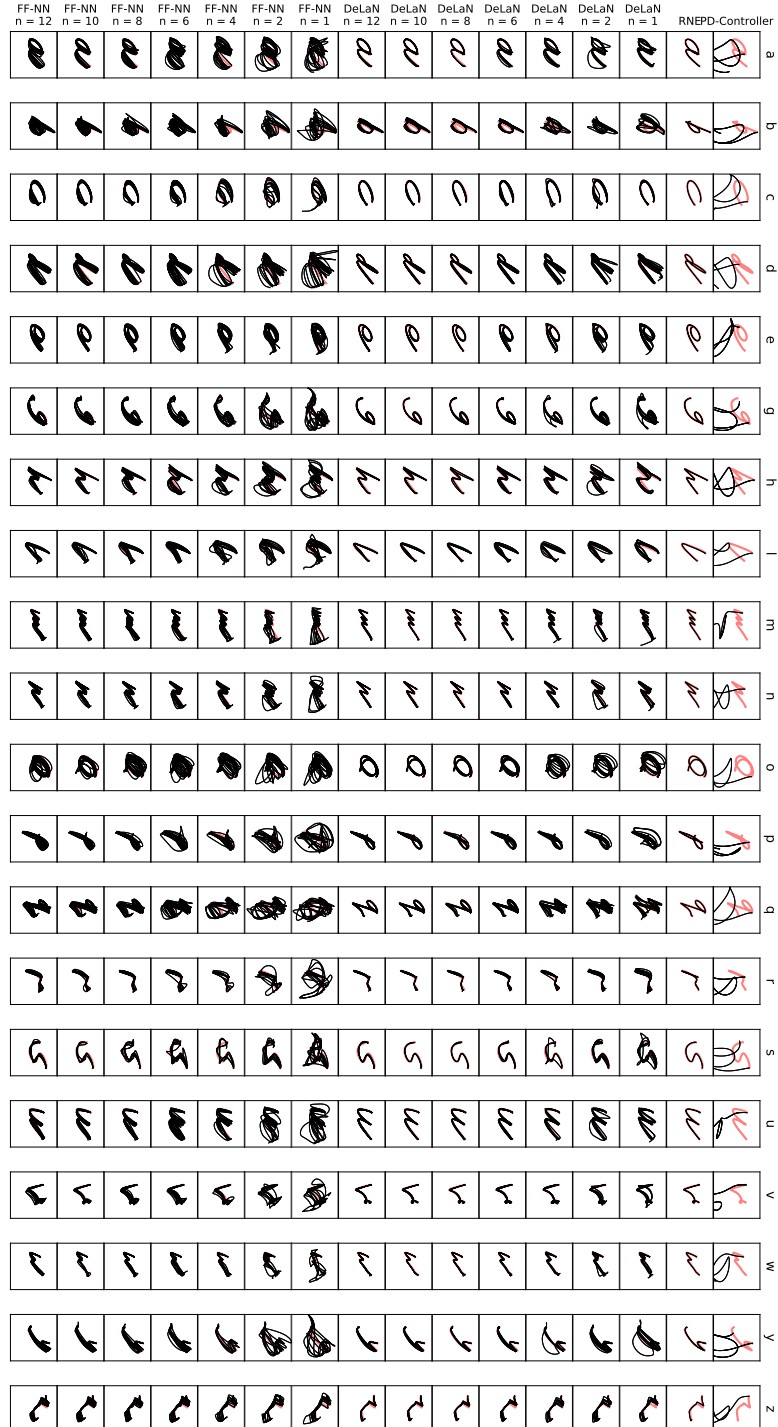

Figure 10: The qualitative performance for the analytic baselines, the feed forward neural network and DeLaN for different number of random training characters. The desired trajectories are shown in red.

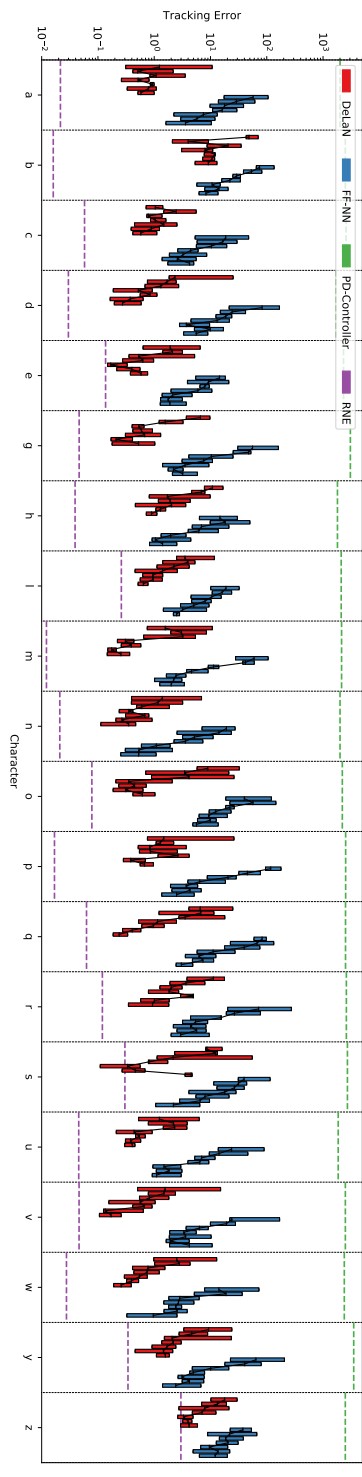

Figure 11: The average performance of DeLaN and the feed forward neural network for each character. The columns of the boxplots correspond to different numbers of training characters, i.e., $n = 1, 2, 4, 6, 8, 10, 12$.

