# OpenReview forum: "Deep Lagrangian Networks: Using Physics as Model Prior for Deep Learning"
_ICLR.cc/2019/Conference_

### Official Review · AnonReviewer1 · 2018-11-03
**Promising, simple approach to model learning. Some questions regarding generalizability to systems with complex dynamics.**

**Rating:** 7
**Confidence:** 3

**Review:**

I like the simplicity of the approach in this paper (especially compared to very computationally hungry methods such as Deepmind's "Graph Networks as Learnable Physics Engines for Inference and Control"). The fact that the approach allows for online learning is also interesting. I very much appreciate that you tested your approach on a real robot arm!

I have a number of questions, which I believe could help strengthen this paper:
- The decomposition of H into L^TL ensures H is positive definite, however there are no constraints on g (gravity/external forces). How do you ensure the model doesn't degenerate into only using g and ignoring H? In the current formulation g only depends on q, however this seems insufficient to model velocity dependent external forces (e.g. contact dynamics). Please elaborate.
- How would you handle partial observability of states? Have you tried this?
- How would you extend this approach to soft robots or robots for which the dimensionality of the state space is unknown?
- Have you tested your method on systems that are not kinematic chains? How would complex contact dynamics be handled (e.g. legged robots)?
- It would be interesting to see more comparisons with recent work (e.g. Deepmind's).

Some figures (e.g. Figure 6) are missing units on the axes. Please fix.

---

> ### Author Response · Authors · 2018-11-19
> **Thank you for the review and the comments regarding the kinematic structure.**
>
> Thank you for your extensive review. Your question regarding closed-loop kinematics chains sparked interesting discussions yielding additional advantages of the approach. We have fixed the issues you mentioned in the figures.
>
> 1)
> Yes, there are no constraints on the decomposition of the torque. This decomposition is unsupervised and could yield degenerate solutions. From our experience, degenerate solutions are learned if one of the components - either H or g - dominates during initialisation. Tuning the hyperparameters for the initialization, i.e., the variance of the gaussian distribution initializing the weights, one achieves a good decomposition into g and H.
>
> To incorporate external forces, one has two options. First, conservative forces, e.g., joints coupled by springs, can be added to V, i.e., V = V_g + V_p. If the external forces are non-conservative, e.g., contact forces, one must decompose \tau. Commonly \tau is decomposed into \tau = \tau_{friction} + \tau_{actuator} + \tau_{external}. The external forces must be projected to the generalized coordinates using \tau_{external} = J_p^{T} f, where f are the external forces acting on point p and J_p the jacobian.
>
>
> 2)
> This depends on the exact definition of partial observability (PO):
>
> - If one interprets PO as observing the state with noise and no direct sensing of the accelerations, DeLan can learn the dynamics using noisy observations and approximated accelerations using finite differences.
>
> - If one interprets PO as missing sensor measurements of a single generalized coordinate, DeLan will not be able of the learning the dynamics. Furthermore, such partial observability would violate the underlying assumption of Lagrangian Mechanics as the system input does not represent generalized coordinates.
>
> - If one interprets PO as an over constrained observation, i.e. a high-dimensional signal that encodes the low-dimensional state, one could learn a latent space embedding, whereas dynamics in the latent space are described by Lagrangian mechanics.
>
>
> 3)
> As the Euler-Lagrange equation (Eq. 3) applies to vibrations and soft robotics, where the state dimensionality is not finite. One could apply an extension of the current approach to soft robotics. We represent the kinetic energy as T = 1/2 \dot{q} H(q) \dot{q}, which applies to system with finite particles. For soft robotics, one would need to represent the kinetic energy as continuous function. Therefore, the Lagrange Euler equation would not simplify to an ODE and one would need to incorporate the PDE. We are currently exploring this direction and don't see any structural problems.
>
>
> 4)
> Thank you for bringing up the different kinematic structures. The problems of closed-loop kinematics are mainly due to use of the Newton-Euler formalism. In contrast, the Lagrangian Mechanics formalism applies to any non-relativistic multi-particle system with holonomic constraints. As closed-loop kinematics only require holonomic constraints, learning the dynamics of closed-loop kinematics can be achieved with DeLan. Older works [1, 2, 3] used Lagrangian Mechanics to manually derive the dynamics of closed-loop kinematics.
>
> Currently, we are looking for publicly available model files of parallel robots (*.sdf or *.urdf) and try to include the evaluations. Up to now, we were not able to find such robot description. If you are aware of such models, we would appreciate your help.
>
> Regarding the contact dynamics. If one can observe the contact force and the point of contact, one can include the contact forces within the learning (see point (1)). If neither is known the learning would be too ambiguous. However, if one has learned the contact-free dynamics, one can compute the external forces on the end-effector and perform force-control without additional sensors. Using system identification this sensorless force control has been done by Wahrburg et. al. [4].
>
>
> 5)
> Yes, we are definitely planning on exploring this approach in future work. We want to use the forward model for planning and compare the performance to black-box model learning. When using the forward model, we will compare to the recent work from Deepmind and other authors.
>
>
> [1] Miller, K., 1992. The Lagrange-based model of Delta-4 robot dynamics. Robotersysteme, 8, pp.49-54.
>
> [2] Liu, K., Lewis, F., Lebret, G., & Taylor, D., 1993. The singularities and dynamics of a Stewart platform manipulator. Journal of Intelligent and Robotic Systems, 8(3), pp.287-308.
>
> [3] Geng, Z., Haynes, L.S., Lee, J.D. and Carroll, R.L., 1992. On the dynamic model and kinematic analysis of a class of Stewart platforms. Robotics and autonomous systems, 9(4), pp.237-254.
>
> [4] Wahrburg, A., Bös, J., Listmann, K. D., Dai, F., Matthias, B., & Ding, H., 2018. Motor-Current-Based estimation of cartesian contact forces and torques for robotic manipulators and its application to force control. IEEE Transactions on Automation Science and Engineering, 15(2), pp.879-886.

---

### Official Review · AnonReviewer3 · 2018-11-05
**Nice approach, but needs to be situated within the relevant work much better**

**Rating:** 4
**Confidence:** 5

**Review:**

This paper looks at system identification for a multi-link robot based upon combining a neural network with the manipulator equations.  Specifically, the authors propose to model the robot dynamics using the typical manipulator equations, but have a deep neural network parameterize the H(q) and g(q) matrices.  They illustrate that the method can control the systems of a simualted 2-dof robot and real Barrett WAM arm, better than a pure neural network modeling approach, PID control, or an analytic model.

Overall, I think there is a genuinely nice application in this paper, but it's not sufficiently compared to existing approaches nor put in the proper context.  There is a lot of language in the paper about encoding the prior via a PDE, but really what the authors are doing is quite simple: they are doing system identification under the standard robot manipulator equations but using a deep network to model the inertia tensor H(q) and the gravity term g(q).  Learning the parameters that make up H(q) and g(q) is completely standard system identification in robotics, but it's interesting to encode these as a generic deep network (I'm somewhat surprised this hasn't been done before, though a quick search didn't turn up any obvious candidates).  However, given this setting, there are several major issues with the presentation and evaluation, which make the paper unsuitable in its current from.

1) Given the fact that the authors are really just in the domain of system identification and control, there are _many_ approaches that they should compare to.  At the very least, however, the authors should compare to standard system identification techniques (see e.g., Wu et al., "An overview of dynamic parameter identification of robots", 2010, and references therein).  This is especially important on the real robot case, where the authors correctly mention that the WAM arm cannot be expressed exactly by the manipulator equations; this makes it all the more important to try identify system parameters via a data-driven approach, not with the hope of finding the exactly "correct" manipulator equations, but with finding some that are good enough to outperform the "analytical" model that the authors mention.  It's initially non-obvious to me that a generic neural network to model the H and g terms would do any better than some of these standard approaches.

2) A lot of the derivations in the text are frankly unnecessary.  Any standard automatic differentiation toolkit will be able to compute all the necessary derivatives, and for a paper such as this the authors can simply specify the architecture of the system (that they use a Cholesky factorization representation of H, with diagonals required to be strictly positive) and let everything else be handled by TensorFlow, or PyTorch, etc.  The derivations in Sections 4.2 and 4.3 aren't needed.

3) The authors keep referring to the Lagrangian equations as a PDE, and while this is true in general for the actual form here it's just a second order ODE; see e.g. https://en.wikipedia.org/wiki/Lagrangian_mechanics.  Moreover, these are really just the standard manipulator equations for multi-link systems, and can just be denoted as such.

Despite these drawbacks, I really do like the overall idea of the approach presented here, it's just that the authors would need to _substantially_ revise the presentation and experiments in order to make this a compelling paper.  Specifically, if they simply present the method as a system identification approach for the manipulator equations, with the key terms parameterized by a deep network (and compare to relevant system identification approaches), I think the results here would be interesting, even if they would probably be more interesting to a robotics audience rather than a core ML audience.  But as it is, the paper really doesn't situation this work within the proper context, making it quite difficult to assess its importance or significance.

---

> ### Author Response · Authors · 2018-11-19
> **Thank your for the review, but we disagree with some points and hope to clarify these aspects.**
>
> We thank the reviewer for the extensive evaluation. We have updated the paper to precisely differentiate between PDE and ODE and we updated the related work section to explain weaknesses of previous approaches and highlight the differences to existing model learning / system identification (SI) approaches.
>
> In addition, we want to clarify the brought-up points below. If you have further questions, please feel free to ask.
>
> 1)
> SI as described in the textbooks or the survey by Wu et. al. [1] - which is btw. missing key references to state-of-the-art methods such as [2, 3] - is non-trivial and hard for real robots. Our lab has significant experience performing model learning on several robot arms, legged robots and robotic hands. However, using state of the art SI [2,3], we learned dynamics parameters, that did NOT outperform the analytical model of the WAM. Therefore, we only use the analytical model as baseline within the paper. Furthermore, we did evaluate our approach against standard black-box SI methods, as the feed-forward neural network is a standard SI technique, which is also mentioned by Wu. et. al. [1].
>
> In addition, we disagree with your statement, that we did not put our work in the proper context. We related our approach to the extensive research covering model learning. Model learning is much broader than SI, as SI is commonly used to refer to model learning with known basis functions. Therefore, we do not limit our comparison to SI but provide a wider context with model learning. Furthermore, the classic SI described by Atkeson et. al. [4] has many limitations (e.g. not applicable to closed-loop kinematics) and will most likely not infer the actual dynamics parameters. As pointed out by Ting et. al. [2] and Nakanishi et. al. [3], the inferred parameters are not guaranteed to yield a positive definite inertia matrix or satisfy the parallel axis theorem - both aspects are ignored within the proposed survey [1] -. In contrast DeLan is guaranteed to yield a physical plausible model, can be applied to any kinematic structure and does not require any knowledge about kinematics.
>
> I hope we could clarify the problems of standard SI and the consequences for real robot models. Furthermore, we are working to provide empirical data. However, we need to re-implement the features derived by the Newton-Euler formulation as our underlying robotics libraries changed and these features require the computation for all transformations, Jacobians along the complete kinematic chain. If you are aware of a public implementation using URDFs as robot descriptors, please let us know.
>
> 2)
> First, we think that reporting the derivatives is good scientific practice and second the analytic computation of the derivatives is necessary for the real-time application. The usage of automatic differentiation in PyTorch does not allow the computation of the feedforward torque with 200Hz. As pointed out in these discussions (https://discuss.pytorch.org/t/how-to-compute-jacobian-matrix-in-pytorch/14968/7, https://stackoverflow.com/questions/43451125/pytorch-what-are-the-gradient-arguments/47026836) the computation of the partial derivatives w.r.t. to network input does not scale well to high-dimensions. If you would prefer to have these derivations within the Appendix, we can also put the derivations within the Appendix.
>
> 3)
> Sorry for being imprecise with the PDE notation. We have updated the paper to be more precise when the equations are referring to a PDE or an ODE.
>
> We also want to point out that Eq. 4 is NOT "just the standard manipulator equations", Eq. 4 applies to any non-relativistic multi-particle system, which can be described with holonomic constraints. Therefore, the Lagrangian Mechanics formalism is applicable for closed-loop kinematic chains, where the standard Newton-Euler approaches fail. Furthermore, most literature related to manipulator equations ignores the functional dependency between C and H, while DeLan explicitly models this functional dependency. We have updated the description to clarify the differences.
>
> [1]Wu, J., Wang, J. and You, Z., 2010. An overview of dynamic parameter identification of robots. Robotics and computer-integrated manufacturing, 26(5), pp.414-419.
>
> [2] Ting, J. A., Mistry, M., Peters, J., Schaal, S., & Nakanishi, J., 2006. A Bayesian Approach to Nonlinear Parameter Identification for Rigid Body Dynamics. In Robotics: Science and Systems, pp. 32-39.
>
> [3] Nakanishi, J., Cory, R., Mistry, M., Peters, J. and Schaal, S., 2008. Operational space control: A theoretical and empirical comparison. The International Journal of Robotics Research, 27(6), pp.737-757.
>
> [4] Atkeson, C. G., An, C. H., & Hollerbach, J. M., 1986. Estimation of inertial parameters of manipulator loads and links. The International Journal of Robotics Research, 5(3), 101-119.

---

> > ### Author Response · Authors · 2018-11-26
> > **Added the offline benchmark including the suggested system identification approach**
> >
> > Dear Reviewer 3,
> >
> > we have added an offline comparison to the Appendix. "Appendix A Offline Benchmarks" compares the performance of DeLaN to the system identification approach introduced by Atkeson et. al. [1], a feed-forward neural network and the recursive Newton Euler algorithm using an analytic model. For this comparison the models as trained offline and evaluated using the mean squared error (MSE) on the training and test set.
> >
> > We added this comparison to the Appendix as we think that the tracking error computed using online learning is the relevant performance indicator and not the offline MSE. We are currently running the online experiments and we will add the results of the system identification approach to the paper as soon as the results become available.
> >
> > We would be very happy, if you could have another look at these results and let us know how we can further improve the paper.
> >
> > [1] Atkeson, C. G., An, C. H., & Hollerbach, J. M., 1986. Estimation of inertial parameters of manipulator loads and links. The International Journal of Robotics Research, 5(3), 101-119.

---

### Official Review · AnonReviewer2 · 2018-11-08
**Interesting paper on using Lagrangian formulation to speed up learning of robot model**

**Rating:** 7
**Confidence:** 4

**Review:**

This paper discusses learning of robot dynamics models. They propose to learn the mass-matrix
and the potential forces, which together describe the Lagrangian mechanics of the robot. The unknown
terms are parametrized as a deep neural network, with some properties (such as positive definiteness)
hard-coded in the network structure. The experimental results show the learned inverse model being used
as the feed-forward term for controlling a physical robot. The results show that this approach lead to faster
learning, as long as the model accurately describes the system. The paper is well written and seems free
of technical errors. The contribution is modest, but relevant, and could be a basis for further research. Below
are a few points that could be improved:

1) The paper uses the term partial differential equation in a non-standard way. While Eqs. 4/5 contain partial derivatives,
the unknown function is q, which is a function of time only. Therefore, the Lagrangian mechanics of robot arms are seen
as ordinary differential equations. The current use of the PDE terms should be clarified, or removed.
2) It is not made clear why the potential forces are learned directly, rather than as a derivative of the potential energy. Could you discuss the advantages/disadvantages?
3) Somewhat related to the previous point: the paper presents learning of dissipative terms as a challenge for future works. Given that the formulation directly allows to add \dot{q} as a variable in g, it seems like a trivial extension. Can you make clearer why this was not done in this paper (yet).
4) The results on the physical robot arm state that the model cannot capture the cable dynamics, due to being a rigid body model. However, the formulation would allow modelling the cables as (non-linear) massless springs, which would probably
explain a large portion of the inaccuracies. I strongly suggest running additional experiments in which the actuator and joints have a separate position, and are connected by springs. If separate measurements of joint-position and actuator position are not available on the arm, it would still be interesting to perform the experiments in simulation, and compare the
performance on hardware with the analytical model that includes such springs.
5) The choice is made to completely hardcode various properties of the mass matrix into the network structure. It would be possible to make some of these properties softcoded. For instance, the convective term C(q,\dot{q})\dot{q} could be learned separately, with the property  C + C^T = \dot{H} encoded as a soft constraint. This would reduce the demand on computing derivatives online.

---

> ### Author Response · Authors · 2018-11-19
> **Thank you for the extensive review, we have clarified the paper and addressed your questions in the comment**
>
> Thank you for providing such extensive review and raising these important questions. If you have further questions, please feel free to ask.
>
> 1)
> Sorry that we have been imprecise on the naming convention. We have updated the paper to make the differences clearer. We replaced Lagrange-Euler PDE with Lagrange-Euler equation and highlight that Eq. 3 can be either a PDE or ODE while Eq. 4 is an ODE. After, Eq. 4 we removed the term PDE. Within the related work section, we use the PDE terminology if the references refer to PDEs.
>
>
> 2)
> Thank you for bringing up this point as this sparked further discussions and new research questions. Until now we have learned the potential forces g(q) directly as this is standard in robotic applications and for our experiments U(q) is not required. However, if one learns dU/dq and U simultaneously one could extend the cost-function with energy conservation and derive energy-based controllers.
>
> A quick offline verification on the simulated WAM data showed that learning dU/dq is possible but currently achieves lower performance. Right now, we cannot conclude if this lower performance is due to hyperparameter settings. Therefore, we are running hyperparameter sweeps to compare the differences.
>
>
> 3)
> Yes, one could just incorporate \dot{q} within g and let g model a mixture of gravity and friction. However, this would contradict Lagrangian Mechanics, where g is the derivative of a E_pot. We would add friction by decomposing \tau into the subparts \tau = \tau_{motor} + \tau_{external} + \tau_{friction}. Simple friction models can be added using the Rayleigh-dissipation function (https://en.wikipedia.org/wiki/Rayleigh_dissipation_function). However, as described by Albu-Schäfer [1] a "good" friction model for robots is described by \tau_{friction} = f(\q, \qd, \tau). Adding such friction model to Lagrangian mechanics is non-trivial. Especially, due to the torque dependency, the computation of the inverse dynamics is challenging. Therefore, incorporating friction would require answering the questions what a sufficiently good friction model is and require an extensive empirical comparison of multiple friction models, which would beyond the scope of this paper.
>
>
> 4)
> Yes, thank you for bringing this to our attention. We agree that performing such experiments would be interesting. Especially for robots with adaptive stiffness as by Braun et. al. [2]. The Barrett WAM does not provide separate motor (\theta) and joint (q) positions. We are looking into performing such experiments in simulation. Below we discuss the theoretic complexity, practical relevance for the performed experiments and implementation difficulties.
>
> From a theoretical perspective including this within the learning should not be too difficult. Rather than learning f^{-1}(q, qd, qdd) = \tau one would learn f^{-1}(q, qd, qdd) - K (\theta - q) = 0 where K is a diagonal matrix with positive entries. The structure of DeLan could be easily adapted for this. We will try to also show this in a small experiment.
>
> From a practical perspective using this model within the controller is more complex. As described in Equation 13.16 in the Springer Handbook of Robotics [3], the inverse model contains d^3 H/d^3t, d^4q/d^4t, d^2g/d^2t, d^2 (d(qd^T H qd)/ dq )/d^2t etc.. Therefore, one would need to compute the higher-order derivatives, which will cause numerical issues that in our opinion would do more harm than help. However, we definitely agree that such model would help when planning using forward model.
>
> From the simulation perspective, we are using PyBullet. To the best of our knowledge, one cannot simulate coupled joints with  Pybullet. Therefore, one has two options. First, simulate the spring outside of PyBullet but this would risk the divergence of the integration of \qdd (PyBullet) and \theta_dd (non-PyBullet). Second one could replace PyBullet with MuJoCo, which can simulate coupled joints but this would require significant implementation effort. Currently, we are evaluating the effort of both.
>
>
> 5)
> Yes, one could add this soft constraint as penalty term. However, the computational overhead of the derivatives is minimal. The derivative computation (Section 4.2 & 4.3) only requires one clamping operation (for the ReLU non-linearity) and one matrix multiplication per hidden layer. This overhead does not hinder the real-time computations and hence, we prefer to use the hard constraint rather than the soft-constraint.
>
>
> [1] Alin Albu-Schäffer. Regelung von Robotern mit elastischen Gelenken am Beispiel der DLRLeichtbauarme. PhD thesis, Technische Universität München, 2002.
>
> [2] Braun, D. J., Howard, M., & Vijayakumar, S., 2012). Exploiting variable stiffness in explosive movement tasks. Robotics: Science and Systems VII, 25.
>
> [3] Siciliano, B., & Khatib, O. (Eds.)., 2016. Springer Handbook of Robotics. Springer.

---

### Comment · Area_Chair1 · 2018-11-19
**further author feedback, or reviewer thoughts after reading the other reviews?**

Thank you to the reviewers for the extensive reviews.
Authors:  We welcome a response.
Reviewers:  We do have high variation in the overall paper evaluation, as reflected in the assigned scores.   Do the other reviews change your evaluation of the paper?

-- your area chair

---

### Comment · Area_Chair1 · 2018-12-08
**reviewers -- comments on author's revisions?**

We are reaching the end of the discussion period.
There remain some mixed opinions on the paper.

The authors have provided detailed replies.
Any further thoughts from the reviewers, in response to those?

Stating pros + cons and summarizing any changes in opinion would be greatly appreciated.

We acknowledge that reviewer & author time is limited.
-- area chair

---

> ### Comment · AnonReviewer1 · 2018-12-08
> **In favor of acceptance, because it's a good first step.**
>
> I've discussed this paper in a reading group with colleagues (without mentioning that I was reviewing it) to get some more opinions and to discover potential flaws.
> The general sentiment was that this method can be difficult to apply in practice, because it has stringent requirements that can be hard to meet with real, systems (e.g. a legged robot). The results show only minor improvements over PD controllers and inverse dynamics controllers, however this might be due to the simplicity of the experiments (2D robot arm).
> That being said, the paper is certainly a step in the right direction and I'm in favor of accepting this paper. The method is sound and simple and the authors present hardware and simulation results. It's a simple framework for others to build upon.

---

> ### Comment · AnonReviewer2 · 2018-12-10
> **Stand by original evaluation**
>
> Neither the comments of the other reviewers nor the response of the authors gives me reason to change my evaluation.
>
> The paper is a small but interesting idea step towards implementing a physics-prior in model learning. Future work should focus on implementing this approach for more complex system, and find out how to scale this approach.
>
> Slight remark: as pointed out in my earlier review, the cable-systems do not violate the physics prior. While I understand that it was not possible to run the suggested new experiments, the error in the text should be corrected.

---

> > ### Author Response · Authors · 2018-12-11
> > **We will update the section about the cable system.**
> >
> > Once we can update the paper, we will make this statement clearer and include the modelling as flexible joint, i.e., as two joint coupled by a massless spring. Furthermore, we will also include that this is not possible with the Barrett WAM as one cannot sense the motor positions.
> >
> > Thanks for bringing this to our attention.

---

### Meta-Review · Area_Chair1 · 2018-12-15
**lean in favor;  one reviewer who lacked time to further evaluate author responses due to failed email notification**

**Confidence:** 4
**Recommendation:** Accept (Poster)

**Metareview:**

The paper looks at a novel form of physics-constrained system identification for a multi-link robot,
although it could also be applied more generally.  The contributions is in many simple; this is seen
in a good light (R1, R3) or more modestly (R2). R3 notes surprise that this hasn't been done before.
Results are demonstrated on a simualted 2-dof robot and real Barrett WAM arm, better than a pure
neural network modeling approach, PID control, or an analytic model.

Some aspects of the writing needed to be addressed, i.e., PDE vs ODE notations.
The point of biggest concern is related to positioning the work relative to other system-identification
literature, where there has been an abundance of work in the robotics and control literature.
There is no final consensus on this point for R3;  R3 did not receive the email notification of the author's detailed reply,
and notes that the author has clarified some respects, but still has concerns, and did not have time to further
provide feedback on short notice.

In balance, the AC believes that this kind of constrained learning of models is underexplored, and
notes that the reviewers (who have considerable shared expertise in robotics-related work) believe
that this is a step in the right direction and that it is surprising this type of approach has not
been investigated yet.  The authors have further reconciled their work with earlier sys-ID work, and
can further describe how their work is situated with respect to prior art in sys-ID (as they do in
their discussion comments).  The AC recommends that: (a) the abstract explicitly mention "system
identification" as a relevant context for the work in this paper, given that the ML audience should
be (or can be) made aware of this terminology; and (b) push more of the math related to the
development of the necessary derivatives to an appendix, given that the particular use of the
derivations seems to be more in support of obtaining the performance necessary for online use,
rather than something that cannot be accomplished with autodiff.